# Nomograms Combining PHI and PI-RADS in Detecting Prostate Cancer: A Multicenter Prospective Study

**DOI:** 10.3390/jcm12010339

**Published:** 2023-01-01

**Authors:** Yongheng Zhou, Qiang Fu, Zhiqiang Shao, Keqin Zhang, Wenqiang Qi, Shangzhen Geng, Wenfu Wang, Jianfeng Cui, Xin Jiang, Rongyang Li, Yaofeng Zhu, Shouzhen Chen, Benkang Shi

**Affiliations:** 1Department of Urology, Qilu Hospital of Shandong University, Jinan 250012, China; 2Department of Urology, Shandong Provincial Hospital, Jinan 250000, China; 3Department of Urology, Linyi People’s Hospital, Linyi 276000, China; 4Department of Thoracic Surgery, Qilu Hospital of Shandong University, Jinan 250012, China

**Keywords:** multiparametric magnetic resonance imaging, prostate health index, prostate cancer, nomogram, diagnosis

## Abstract

(1) Background: The study aimed to construct nomograms to improve the detection rates of prostate cancer (PCa) and clinically significant prostate cancer (CSPCa) in the Asian population. (2) Methods: This multicenter prospective study included a group of 293 patients from three hospitals. Univariable and multivariable logistic regression analysis was performed to identify potential risk factors and construct nomograms. Discrimination, calibration, and clinical utility were used to assess the performance of the nomogram. The web-based dynamic nomograms were subsequently built based on multivariable logistic analysis. (3) Results: A total of 293 patients were included in our study with 201 negative and 92 positive results in PCa. Four independent predictive factors (age, prostate health index (PHI), prostate volume, and prostate imaging reporting and data system score (PI-RADS)) for PCa were included, and four factors (age, PHI, PI-RADS, and Log PSA Density) for CSPCa were included. The area under the ROC curve (AUC) for PCa was 0.902 in the training cohort and 0.869 in the validation cohort. The AUC for CSPCa was 0.896 in the training cohort and 0.890 in the validation cohort. (4) Conclusions: The combined diagnosis of PHI and PI-RADS can avoid more unnecessary biopsies and improve the detection rate of PCa and CSPCa. The nomogram with the combination of age, PHI, PV, and PI-RADS could improve the detection of PCa, and the nomogram with the combination of age, PHI, PI-RADS, and Log PSAD could improve the detection of CSPCa.

## 1. Introduction

Globally, prostate cancer (PCa) is the second leading cause of death among men, with approximately 268,490 new cases and 34,500 deaths projected to occur in America by 2022. [1]. With the widespread use of prostate-specific antigen (PSA), the early diagnosis and treatment of PCa are gradually increasing [2]. However, the low specificity of PSA has led to lots of unnecessary and excessive prostate biopsies, resulting in a significant financial burden as well as many post-biopsy complications. In recent years, scholars have used different biomarkers, such as the 4Kscore, PCA3, and the prostate health index (PHI), and different predictive models to improve the detection rate of prostate cancer [3,4,5]. The clinical application of prostate multiparametric magnetic resonance imaging (mpMRI) and the prostate imaging reporting and data system (PI-RADS) has also improved the diagnosis of PCa and clinically significant prostate cancer (CSPCa) in terms of imaging [6]. With the combination of the above biomarkers with mpMRI, cancer detection rates have been improved and unnecessary biopsies have been reduced [7].

The discovery and clinical application of (-2) proPSA (P2PSA) have made PHI an important indicator for low-risk and intermediate-risk PCa screening, especially in PSA 2–20 ng/mL, in clinical practice [8,9]. A large cohort study showed that a cutoff value of 35 for PHI in Asian populations reached good sensitivity and specificity [8]. However, in actual clinical work, it is insufficient to use the PHI value of 35 as a cutoff value for diagnosing prostate cancer. Therefore, the role of the combined diagnosis of PCa appears to be important.

The purpose of this study is to construct clinically useful nomograms using PHI and PI-RADS indicators, along with other clinical indicators, which are based on data from a multicenter database, in order to improve the diagnostic accuracy of PCa and CSPCa in the Asian population.

## 2. Materials and Methods

### 2.1. Study Population

This multicenter prospective study included a group of 293 patients from three hospitals in the Asian population, 29 patients from hospital 1, 42 patients from hospital 2, and 222 patients from hospital 3. This study is a prospective multicenter observational cohort study and the clinical trial registration number is NCT05179707. It has been approved by the Ethics Committee of Qilu Hospital of Shandong University and endorsed by the Ethics Committees of the other institutions participating in the study. All patients signed a written informed consent form. Patients with PSA in 4–20 ng/mL and a normal digital rectal examination were enrolled. If a patient’s mpMRI showed a low probability of cancer and a PSA level of around 4 ng/mL, we elaborated on different treatment options for the patient, including active surveillance and other treatment modalities. If the patient had a very strong desire for a biopsy, we performed a biopsy after that patient signed an informed consent form. 

The exclusion criteria were as follows: (I) abnormal blood clotting function; (II) infection of the urinary tract or prostatitis; (III) prostate surgery (such as transurethral resection of the prostate) performed prior to biopsy. The patients in this cohort were all biopsy-naive. 

### 2.2. Data Collection and Clinical Variables

Before prostate biopsy, blood samples were collected prospectively to determine total prostate-specific antigen (TPSA), free prostate-specific antigen (fPSA), and P2PSA levels. A blood clotting process was performed at room temperature for one hour, followed by centrifugation for fifteen minutes. A serum sample was aliquoted, frozen at −80 °C, and subjected to immunoassay using dedicated Access TPSA, fPSA, and P2PSA reagents (Beckman Coulter, Brea, CA, USA). Calculation of the f/T indicator was completed by dividing the fPSA by the TPSA, and calculation of the PSAD was performed by dividing the TPSA by the PV. These data were calculated using the prostate ellipsoid formulation: PV = ([maximum anteroposterior diameter] × [maximum transverse diameter] × [maximum longitudinal diameter] × 0.52], measured using an MRI scan [10]. Based on Beckman and Coulter’s PHI formula, the PHI was calculated as follows: ((-2) proPSA/free PSA) / √PSA, and %P2PSA was calculated using the formula [(P2PSA pg/mL)/ (fPSA ng/mL × 1000)] × 100 [9,11]. 

A mpMRI was performed on all patients prior to prostate biopsy using a 3.0 T machine without an endorectal coil. The scanning protocol of mpMRI included T1-weighted imaging (T1WI), T2-weighted imaging (T2WI), diffusion-weighted imaging (DWI), and dynamic contrast-enhanced imaging (DCE). DWI was acquired with b values of 0 and 1500 s/mm^2^, and an apparent diffusion coefficient (ADC) map was generated. The mpMRI was interpreted by two urogenital radiologists with at least three years of experience in prostate MRI and recorded by using the PI-RADS v2.1 score. There is a very low probability that CSPCa will be present in PI-RADS 1 (CSPCa is highly unlikely to occur); PI-RADS 2 (CSPCa is highly unlikely to occur); PI-RADS 3 (equivocal presence of CSPCa); PI-RADS 4—High (CSPCa is highly likely to occur); PI-RADS 5—Very high (CSPCa is highly likely to occur) [12,13].

All patients underwent ultrasound-guided transperineal prostate biopsy or transrectal prostate biopsy in antibiotic prophylaxis. The patients underwent 12-core systematic prostate biopsy and an additional 4-core biopsy was performed in suspicious lesions. MRI-transrectal/transperineal cognitive fusion biopsy was performed for the suspicious lesions. When using transperineal prostate biopsy, physicians use a free-hand approach biopsy. Biopsies were performed at each center by physicians with at least five years of experience in biopsy procedures. According to the guidelines of the International Society of Urological Pathology Consensus Conference, biopsy specimens were interpreted and graded [14]. PCa was defined as Gleason score (GS) ≥ 3 + 3 and CSPCa was defined as GS ≥ 3 + 4 [15].

### 2.3. Construction of the PCa and CSPCa Nomograms

The entire cohort was randomly divided into a training cohort and a validation cohort in a 3:1 ratio, and we used the training cohort to build the nomogram and the validation cohort for verification. The potential risk factors for PCa and CSPCa were identified using a univariable logistic regression analysis. The factors with a P value less than 0.1 in univariable logistic regression analysis were included in the multivariable logistic regression analysis. The final predictive models using the independent risk factors (*p* < 0.05 in multivariable stepwise forward logistic regression) were constructed. Following the multivariable logistic regression analysis, nomograms were constructed using the R packages “rms” and “DynNom” (version 4.1.1; http://www.r-project.org/, 3 August 2022). Using the regression model, scores were calculated for each variable, and the predicted probability of PCa and CSPCa was determined by averaging the scores.

### 2.4. Nomogram Performance

In order to evaluate the performance of the nomogram, discrimination, calibration, and clinical utility were taken into account. Discrimination consists of evaluating a model for its ability to distinguish between events and non-events. An evaluation of the predictive nomogram’s discrimination efficiency was conducted using a receiver operating characteristic (ROC) curve [16]. A calibration process was used to determine the degree to which predicted probabilities correspond to actual results. The calibration power was assessed using the Hosmer–Lemeshow test, and a P value greater than 0.05 was considered satisfactory. A bootstrapping method with 1000 replications was used for internal validation [17]. Evaluation of clinical utility was conducted using decision curve analysis (DCA). 

### 2.5. Statistical Analysis

For the comparison of the continuous variables of groups, the normality test was first performed, and the Student t-test was used for continuous variables that met the normality test; otherwise, the Mann–Whitney U test was applied for continuous variables. Normally distributed continuous variables were described as mean ± standard deviation (SD); otherwise, the form of the median (interquartile range (IQR)) was described. Ranked data were analyzed by using the Wilcoxon rank sum test. The Kruskall Wallis test was used to analyze the variables between multiple groups. Some indicators with over-inflated odds ratio (OR) values were balanced using Log transformation. The optimal cut-off value of the nomogram was obtained from the maximum Youden index. *p* value < 0.05 was considered statistically significant. Data analysis was conducted using R Project software (version 4.1.1; http://www.R-project.org, 3 August 2022) and SPSS software (version 25.0; SPSS Inc., Chicago, IL, USA).

## 3. Results

A total of 293 patients were included in our study with 201 negative and 92 positive results in PCa between September 2020 to June 2022. A comparison of the baseline demographic characteristics from the three hospitals is shown in Appendix A. In the cohort, patients were randomly assigned to the training cohort (n = 220) or the validation cohort (n = 73). No significant differences were observed in any of the variables between the two cohorts (Table 1). The characteristics of patients in the training and validation cohorts are shown in Table 2 and Table 3.

### 3.1. Univariable and Multivariable Regression Analyses in Predicting PCa and CSPCa

An evaluation of the risk factors for PCa and CSPCa in the training cohort was conducted using both univariable and multivariable stepwise forward regression analyses (Table 4). Univariable logistic regression analyses showed that age, TPSA, P2PSA, PHI, f/T, %P2PSA, PV, PI-RADS, and Log (PSAD) were risk factors in predicting PCa and CSPCa. After analysis of the clinical value of the predictors and the collinearity, age, TPSA, PHI, f/T, PV, and PI-RADS were included into the multivariable regression analysis. Multivariable stepwise forward regression analysis revealed that age (OR = 0.970; 95% confidence interval (CI): 0.952–0.988; *p* = 0.014), PHI (OR = 1.037; 95% CI: 1.022–1.052; *p* = 0.000), PV (OR = 0.970; 95% CI: 0.952–0.988; *p* = 0.002), and PI-RADS (OR = 2.936; 95% CI: 1.873–4.601; *p* = 0.000) were predictive factors in detecting PCa. The risk factors for detecting CSPCa in multivariable regression analysis were PHI (OR = 1.033; 95% CI: 1.020–1.045; *p* = 0.000), Log (PSAD) (OR = 9.758; 95% CI: 2.458–39.220; *p* = 0.001), and PI-RADS (OR = 2.458; 95% CI: 1.709–3.535; *p* = 0.000).

### 3.2. The Construction and Performance of Nomogram

Four independent predictive factors (age, PHI, PV, and PI-RADS) for PCa were included and four factors (age, PHI, PI-RADS, and Log PSAD) for CSPCa were included. Detailed information on the predictive model is shown in Table 5. The predictive models of PCa and CSPCa were constructed based on coefficients of the multivariable logistic regression model and are shown in Figure 1. There were totals of 7 axes in this nomogram, and 4 axes represented predictive factors. In order to calculate the estimated score for each risk factor, a perpendicular line can be drawn along the axis of the top points, and an additional sum can be computed to determine the total score. Additionally, we developed two web-based operation interfaces (https://zhouyonghengql.shinyapps.io/PCa_DynNom/) (https://zhouyonghengql.shinyapps.io/CSPCa_DynNomapp/) using the “Dynnom” package for urology surgeons in order to facilitate the widespread use of our predictive nomograms on 20 August 2022.

The ROC curve was used to evaluate the accuracy of the predictive models and nomograms in discrimination capacity (Figure 2). The area under the ROC curve (AUC) for PCa was 0.9023 (95% CI: 0.8578–0.9467) in the training cohort and 0.8690 (95% CI: 0.7673–0.9707) in the validation cohort, which indicated that the nomogram had relatively high predictive accuracy. The optimal cut-off of the nomogram was 0.304, and the specificity and sensitivity were 0.841 and 0.859, respectively. In addition, the nomogram could avoid 57.68% of biopsies, and only 4.44% of patients with PCa were missed in this cut-off value. 

The Hosmer–Lemeshow test and calibration plot were used to assess calibration power. According to the Hosmer–Lemeshow test, the P value in the training cohort was 0.084 and, in the validation cohort, it was 0.397, indicating that the difference between the predicted probabilities and the actual probabilities was not significant. Both the training and validation cohort calibration plots (Figure 3) demonstrate that the predictive nomogram was well-calibrated. The DCA curve is shown in Appendix A. 

The different cut-off values of PHI and the optimal cut-off values of nomograms are shown in Table 6. When the PHI value was greater than or equal to 35, the sensitivity and the specificity were 95.77% and 34.90%, respectively, and 23.64% of biopsies could be saved. When applying the nomogram for predicting PCa, 55.91% of biopsies could be saved, accompanied by 3.67% of PCa as well as 1.82% of CSPCa being missed.

## 4. Discussion

PCa is one of the common malignant tumors in men and prostate biopsy remains the gold standard for confirming PCa [18]. However, many patients experience unnecessary biopsies and suffer from the complications of biopsies. Therefore, the combined diagnosis of PCa has become quite important. Hsieh et al. found that the AUC of the combination of PHI and mpMRI (0.873 (95% CI 0.8050–0.9407)) was higher than the AUC of the PHI (0.735 (95% CI 0.6194–0.8497)) and the AUC of the mpMRI (0.830 (95% CI 0.7598–0.9004)) [19]. Other scholars also explored and constructed many different combined models to improve the diagnostic accuracy of PCa [7,19,20,21,22]. 

It is well known that mpMRI is gradually spreading in the diagnostic application of PCa [23]. There are a lot of authors that have studied it and have offered interesting results in this regard. Grey et al. derived the negative predictive value of 97.7% for the PI-RADS score in the diagnosis of CSPCa [24]. They thought the PI-RADS scoring could be used in the decision-making process for detecting CSPCa. A systematic review from the Cochrane Database illustrated the benefit of detecting more CSPCa in mpMRI-targeted biopsies with a sensitivity of 0.80 (95% CI: 0.69–0.87) and a specificity of 0.94 (95% CI: 0.90–0.97) [25]. Mendhiratta et al. reported that targeted biopsy based on the mpMRI could detect more CSPCa than systematic biopsy (88.6% vs. 77.3%, *p*=0.037), which reflected the strong predictive efficiency of mpMRI in CSPCa [6]. The clinical application of mpMRI and the criteria for PI-RADS scoring are described in the ESUR prostate MR guidelines, providing clinicians with further improvements in the learning of mpMRI as well [26].

In this study, we developed clinical prediction models and devised nomograms using the combination of PHI, PI-RADS scores, and other important clinical predictors and developed a website that promotes our nomograms. For patients with elevated PSA but low predictive probability, measures such as active monitoring can be used.

Prostate biopsy is already a routine procedure and can be performed in many hospital outpatient operating rooms. With the widespread of transperineal prostate biopsy techniques, complications such as sepsis have decreased [27]. However, in some elderly patients with other diseases or poor coagulation function, prostate biopsy under local anesthesia still carries a high risk of bleeding. Therefore, a clinical predictive tool should be used to determine whether to perform active monitoring or to perform biopsy under close supervision. 

Prior studies have constructed a number of nomograms that incorporate PHI and other clinical risk factors or PI-RADS and other clinical risk factors for PCa or CSPCa [20,21,22]. The superiority of the combined diagnosis of PHI and PI-RADS has also been demonstrated in several studies [19,28]. However, no studies constructed nomograms with the combination of PHI, PI-RADS scores, and other clinically significant predictive factors. Considering previous studies and the usefulness as well as the convenience of a clinical predictive model, we included four independent predictive factors in detecting PCa: age, PHI, PI-RADS, and PV. In predicting the positive rate of CSPCa, four predictive factors were included: age, PHI, PI-RADS, and Log PSAD. Although age had a P value of 0.084 for PCa in the univariable regression analysis, we still decided to include age in the model because age has been clinically identified as a risk factor in the development of PCa [29]. According to several observational studies, the diagnosis of patients with older age for PCa is associated with a poor prognosis [30,31]. As the (-2) proPSA was found in 1997, PHI is gradually becoming an effective means of screening for PCa [32] and has shown good AUC in detecting PCa and CSPCa [9,33]. As mentioned above, the nomogram studied in this study is more applicable to patients with TPSA between 4 and 20 ng/mL who are able to undergo the PHI test as well as the mpMRI examination. Although the applicability conditions are more stringent, it is beneficial to increase the detection rate of patients in this TPSA interval.

There are many previous nomograms for predicting PCa and studies combining PHI and PI-RADS score for detecting PCa [19,22]. Although the benefits of combining PHI with mpMRI are well recognized, the nomogram combining PHI with mpMRI has not been studied. As compared to previously published PCa and CSPCa predictive models, our study offers the following advantages. First, we visualized the prediction model as nomograms and developed a website with an operation interface for our nomogram on 20 August 2022, (https://zhouyonghengql.shinyapps.io/PCa_DynNom/), (https://zhouyonghengql.shinyapps.io/CSPCa_DynNomapp/), which greatly improved in terms of efficiency, accuracy, and clinical usability as a result of this optimization. Secondly, the combination of serum-specific biomarkers PHI and mpMRI also enables the combined diagnosis of physiological and anatomical functions, which can reduce the number of unnecessary biopsies by more than half.

It is worth mentioning that in our study, we analyzed the sensitivity and specificity of different cutoff values of PHI, and we found that as the cutoff value of PHI increased, the missed PCa and CSPCa also increased gradually. However, for the cut-off value of PHI of 35 [8], which is commonly used in clinical practice, our study found that its specificity is low, and it is necessary to appropriately increase the threshold of PHI for the detection of cancer. When the prediction rate for PCa by the nomogram is greater than 27%, our study suggests that prostate biopsy should be performed in this population with a low risk of missing CSPCa.

The following limitations were also included in our study. First, although this study is a prospective multicenter cohort study, the population sample size of our study was small, which may have some limitations. Secondly, there are many clinical studies that are still controversial and have not reached a consensus on the definition of CSPCa, and the GS ≥ 3 + 4 seems to be prevalent in most recent criteria [15,34]. We, therefore, used the definition in our study. In addition, maximum core length was used in the definition of CSPCa; however, we did not incorporate it into the final analysis, as it was not available for all patients. The use of a nomogram in this study can predict the probability of developing CSPCa before biopsy and can provide good treatment advice to patients. However, this study did not correlate the predictive results of the nomogram with the risk of CSPCa at the time of radical prostatectomy or the risk of adverse pathological features of radical prostatectomy, which remains a direction for future research and has considerable clinical implications. Finally, a larger sample and external validation are still needed to prove our conclusions and update our nomograms.

## 5. Conclusions

The combined diagnosis of PHI and PI-RADS can avoid more unnecessary biopsies. The nomogram with the combination of age, PHI, PV, and PI-RADS could improve the detection of PCa, and the nomogram with the combination of age, PHI, PI-RADS, and Log PSAD could improve the detection of CSPCa.

## Figures and Tables

**Figure 1 jcm-12-00339-f001:**
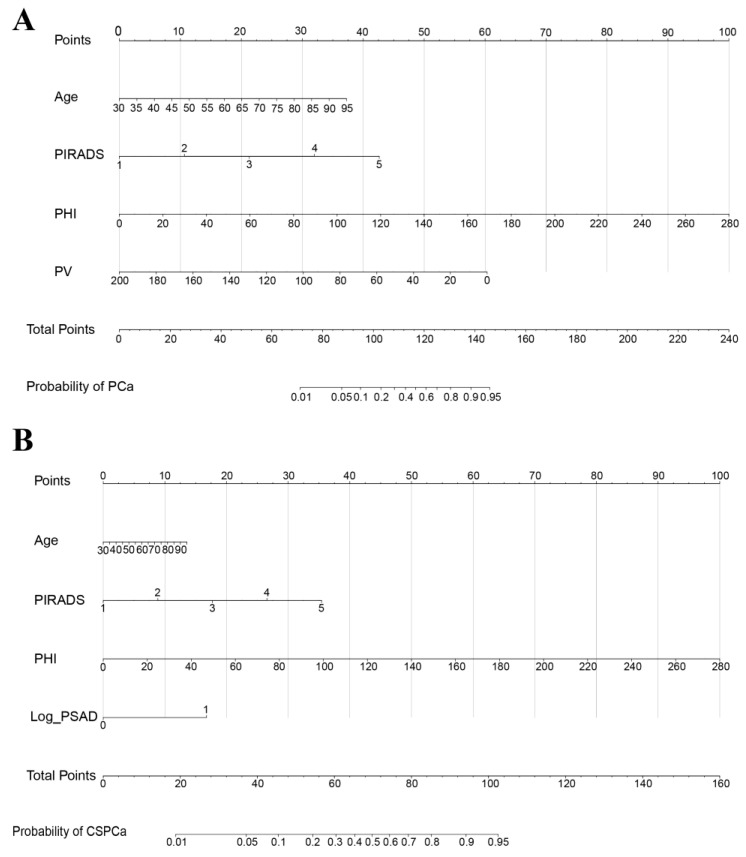
Nomograms for PCa (**A**) and CSPCa (**B**). In order to determine the point of each variable, draw a vertical line from the corresponding axis of the variable to the points axis. To estimate the probability of PCa/CSPCa, the total score can be projected to the lower total point axis by summing the points for each variable. PIRADS: Prostate Imaging-Reporting and Data System; PHI: prostate health index; PV: prostate volume; PSAD: prostate-specific antigen density; PCa: prostate cancer; CSPCa: clinically significant prostate cancer, defined as Gleason Grade ≥ 2.

**Figure 2 jcm-12-00339-f002:**
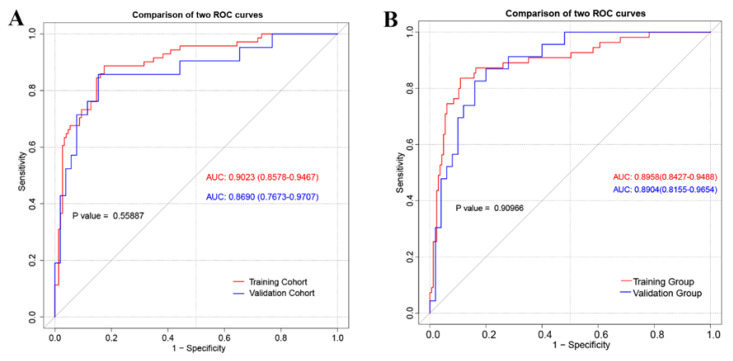
The receiver operating characteristic (ROC) curve of training cohort and validation cohort for PCa (**A**) and CSPCa (**B**). PCa: prostate cancer; CSPCa: clinically significant prostate cancer, defined as Gleason Grade ≥ 2; AUCs, areas under the ROC curve.

**Figure 3 jcm-12-00339-f003:**
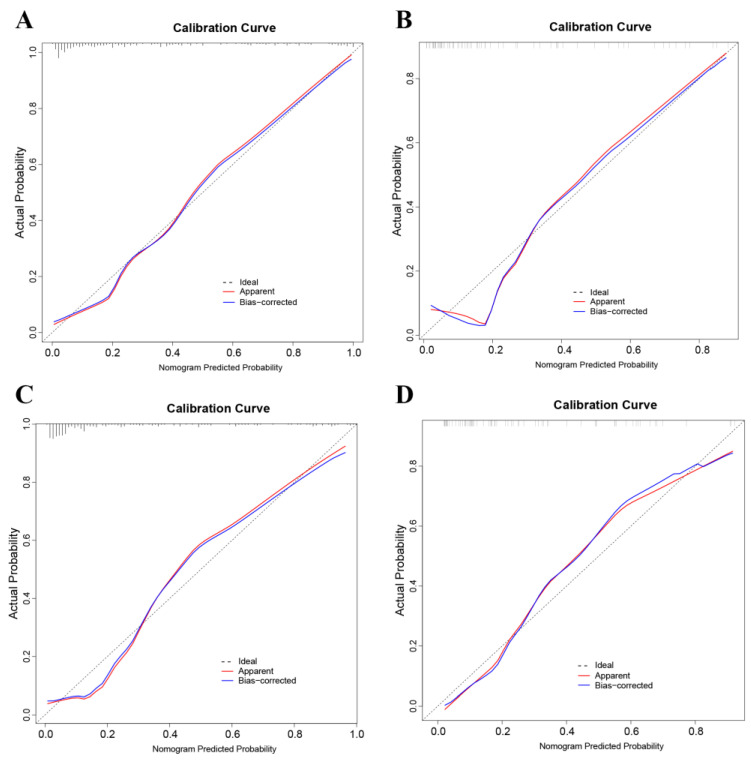
Prediction nomogram calibration curves for PCa in the training cohort (**A**) and validation cohort (**B**). The calibration curves for the CSPCa prediction nomogram in the training cohort (**C**) and validation cohort (**D**). On the x-axis, the nomogram-predicted probability is displayed, while on the y-axis, the actual probability of PCa or CSPCa is displayed. An ideal curve with a black point is represented by the black pointed line, an apparent curve with a red solid line represents the apparent curve that has not been corrected, and a bias-correction curve derived from bootstrapping (B = 1000 repetitions) is represented by the blue solid line. PCa: prostate cancer; CSPCa: clinically significant prostate cancer, defined as Gleason Grade ≥ 2.

**Table 1 jcm-12-00339-t001:** Patients’ characteristics of the training cohort and validation cohort in total and significant prostate cancer.

Characteristics	All Cohort	PCa			CSPCa		
		Training Cohort	Validation Cohort	*p* Value	Training Cohort	Validation Cohort	*p* Value
N (%)	293 (100)	220 (75.09)	73 (24.91)	-	220 (75.09)	73 (24.91)	-
Age (years), median (IQR)	66.00 (60.00–72.00)	66.00 (59.25–72.75)	66.00 (61.00–72.00)	0.787	66.00 (60.00–72.00)	66.00 (60.00–74.00)	0.355
TPSA (ng/mL), median (IQR)	8.51 (5.97–12.11)	8.59 (5.96–12.13)	8.31 (5.95–11.93)	0.956	8.51 (5.88–11.96)	8.84 (6.11–12.99)	0.478
fPSA (ng/mL), median (IQR)	1.13 (0.79–1.61)	1.12 (0.79–1.60)	1.14 (0.76–1.73)	0.697	1.13 (0.75–1.60)	1.21 (0.91–1.67)	0.396
P2PSA (ng/mL), median (IQR)	17.89 (12.01–28.90)	17.97 (12.95–22.35)	17.89 (10.98–29.76)	0.783	17.83 (11.80–28.62)	20.54 (14.25–29.87)	0.226
PHI, median (IQR)	47.15 (35.36–67.90)	47.65 (35.18–67.91)	46.28 (35.16–68.51)	0.842	46.51 (25.09–69.60)	48.80 (37.42–63.95)	0.574
f/T, median (IQR)	0.14 (0.10–0.19)	0.14 (0.09–0.19)	0.13 (0.11–0.19)	0.690	0.14 (0.09–0.20)	0.14 (0.10–0.19)	0.679
%P2PSA, median (IQR)	1.70 (1.27–2.27)	1.71 (1.30–2.23)	1.70 (1.15–2.28)	0.955	1.69 (1.26–2.27)	1.73 (1.33–2.28)	0.582
PV (mL), median (IQR)	44.13 (28.84–65.54)	44.45 (28.84–66.23)	43.68 (28.53–63.86)	0.820	42.46 (28.22–63.10)	45.45 (31.43–67.27)	0.381
PI-RADS, n (%)				0.963			0.359
≤2	117 (39.9)	85 (38.6)	32 (43.8)		91 (41.4)	26 (35.6)	
3	92 (31.4)	76 (34.5)	16 (21.9)		69 (31.4)	23 (31.5)	
≥4	84 (28.7)	59 (26.8)	25 (34.2)		60 (27.3)	24 (32.9)	
PSAD (ng/mL^2^), median (IQR)	0.19 (0.13–0.31)	0.18 (0.12–0.31)	0.21 (0.13–0.33)	0.589	0.18 (0.12–0.31)	0.21	0.871

IQR: interquartile range; TPSA: total prostate-specific antigen; fPSA: free prostate-specific antigen; P2PSA: (-2)pro-prostate-specific antigen; PHI: prostate health index; f/T: free/total prostate-specific antigen; %P2PSA: defined as [(P2PSA/fPSA) × 100]; PV: prostate volume; PI-RADS: Prostate Imaging-Reporting and Data System; PSAD: prostate-specific antigen density; CSPCa: clinically significant prostate cancer, defined as Gleason Grade ≥ 2. The P value is for comparing the training cohort with the validation cohort.

**Table 2 jcm-12-00339-t002:** Patient characteristics in training and validation cohorts with and without PCa.

Characteristics	Training Cohort			Validation Cohort		
	Non-PCa	PCa	*p* Value	Non-PCa	PCa	*p* Value
Age (years), median (IQR)	66.00 (59.00–71.50)	67.00 (64.00–74.00)	0.094	63.50 (58.00–69.75)	71.00 (66.00–77.00)	0.001
TPSA (ng/mL), median (IQR)	8.38 (5.57–11.59)	8.97 (6.38–13.58)	0.106	7.98 (5.65–10.48)	11.03 (7.14–13.69)	0.023
fPSA (ng/mL), median (IQR)	1.23 (0.81–1.70)	1.06 (0.79–1.39)	0.196	1.11 (0.85–1.69)	1.40 (0.74–1.78)	0.622
P2PSA (ng/mL), median (IQR)	16.57 (10.93–23.62)	25.57 (31.58–52.10)	0.000	15.20 (9.65–24.64)	30.44 (15.45–49.74)	0.001
PHI, median (IQR)	42.07 (31.58–52.10)	72.57 (51.67–110.15)	0.000	40.74 (28.62–53.92)	73.11 (59.45–98.91)	0.000
f/T, median (IQR)	0.15 (0.10–0.21)	0.12 (0.09–0.15)	0.001	0.14 (0.11–0.20)	0.13 (0.08–0.17)	0.338
%P2PSA, median (IQR)	1.47 (1.03–1.85)	2.44 (1.87–3.20)	0.000	1.58 (1.05–2.03)	2.30 (1.76–3.23)	0.000
PV (mL), median (IQR)	50.39 (35.03–73.81)	32.85 (23.06–47.67)	0.000	50.16 (32.90–66.55)	30.40 (20.14–48.64)	0.010
PI-RADS, n (%)			0.000			0.002
≤2	77 (51.7)	8 (11.3)		29 (55.8)	3 (14.3)	
3	51 (34.2)	25 (35.2)		11 (21.2)	5 (23.8)	
≥4	21 (14.1)	38 (53.5)		12 (23.1)	13 (61.9)	
PSAD (ng/mL^2^), median (IQR)	0.16 (0.11–0.24)	0.25 (0.18–0.47)	0.000	0.17 (0.11–0.25)	0.36 (0.23–0.43)	0.000

IQR: interquartile range; TPSA: total prostate-specific antigen; fPSA: free prostate-specific antigen; P2PSA: (-2)pro-prostate-specific antigen; PHI: prostate health index; f/T: free/total prostate-specific antigen; %P2PSA: defined as [(P2PSA/fPSA) × 100]; PV: prostate volume; PI-RADS: Prostate Imaging-Reporting and Data System; PSAD: prostate-specific antigen density; PCa: prostate cancer. P value is for the comparison between non-PCa and PCa in the training cohort and validation cohort, respectively.

**Table 3 jcm-12-00339-t003:** Patient characteristics in training and validation cohorts with and without CSPCa.

Characteristics	Training Cohort			Validation Cohort		
	Non-PCa	PCa	*p* Value	Non-PCa	PCa	*p* Value
Age (years), median (IQR)	66.00 (59.00–71.50)	67.00 (64.00–74.00)	0.094	63.50 (58.00–69.75)	71.00 (66.00–77.00)	0.001
TPSA (ng/mL), median (IQR)	8.38 (5.57–11.59)	8.97 (6.38–13.58)	0.106	7.98 (5.65–10.48)	11.03 (7.14–13.69)	0.023
fPSA (ng/mL), median (IQR)	1.23 (0.81–1.70)	1.06 (0.79–1.39)	0.196	1.11 (0.85–1.69)	1.40 (0.74–1.78)	0.622
P2PSA (ng/mL), median (IQR)	16.57 (10.93–23.62)	25.57 (31.58–52.10)	0.000	15.20 (9.65–24.64)	30.44 (15.45–49.74)	0.001
PHI, median (IQR)	42.07 (31.58–52.10)	72.57 (51.67–110.15)	0.000	40.74 (28.62–53.92)	73.11 (59.45–98.91)	0.000
f/T, median (IQR)	0.15 (0.10–0.21)	0.12 (0.09–0.15)	0.001	0.14 (0.11–0.20)	0.13 (0.08–0.17)	0.338
%P2PSA, median (IQR)	1.47 (1.03–1.85)	2.44 (1.87–3.20)	0.000	1.58 (1.05–2.03)	2.30 (1.76–3.23)	0.000
PV (mL), median (IQR)	50.39 (35.03–73.81)	32.85 (23.06–47.67)	0.000	50.16 (32.90–66.55)	30.40 (20.14–48.64)	0.010
PI-RADS, n (%)			0.000			0.002
≤2	77 (51.7)	8 (11.3)		29 (55.8)	3 (14.3)	
3	51 (34.2)	25 (35.2)		11 (21.2)	5 (23.8)	
≥4	21 (14.1)	38 (53.5)		12 (23.1)	13 (61.9)	
PSAD (ng/mL^2^), median (IQR)	0.16 (0.11–0.24)	0.25 (0.18–0.47)	0.000	0.17 (0.11–0.25)	0.36 (0.23–0.43)	0.000

IQR: interquartile range; TPSA: total prostate-specific antigen; fPSA: free prostate-specific antigen; P2PSA: (-2)pro-prostate-specific antigen; PHI: prostate health index; f/T: free/total prostate-specific antigen; %P2PSA: defined as [(P2PSA/fPSA) × 100]; PV: prostate volume; PI-RADS: Prostate Imaging-Reporting and Data System; PSAD: prostate-specific antigen density; CSPCa: clinically significant prostate cancer, defined as Gleason Grade ≥ 2. P value is for the comparison between non-CSPCa and CSPCa in the training cohort and validation cohort, respectively.

**Table 4 jcm-12-00339-t004:** Univariable and multivariable logistic regression analysis of risk factors for total and significant prostate cancer in the training cohort.

Variable	PCa						CSPCa					
	Univariable Analysis			Multivariable Analysis			Univariable Analysis			Multivariable Analysis		
	OR	95% CI	*p* Value	OR	95% CI	*p* Value	OR	95% CI	*p* Value	OR	95% CI	*p* Value
Age	1.028	0.996–1.061	0.084	0.970	0.952–0.988	0.014	1.038	1.002–1.076	0.040			
TPSA	1.057	0.993–1.124	0.081				1.097	1.024–1.176	0.008			
fPSA	0.736	0.499–1.085	0.122				0.794	0.527–1.196	0.269			
P2PSA	1.045	1.025–1.066	0.000				1.047	1.026–1.068	0.000			
PHI	1.044	1.030–1.059	0.000	1.037	1.022–1.052	0.000	1.046	1.032–1.061	0.000	1.033	1.020–1.045	0.000
f/T	0.002	0.000–0.196	0.007				0.001	0.000–0.078	0.003			
%P2PSA	3.652	2.389–5.583	0.000				3.004	2.058–4.383	0.000			
PV	0.970	0.956–0.984	0.000	0.970	0.952–0.988	0.002	0.964	0.947–0.981	0.000			
PI-RADS	3.385	2.319–4.941	0.000	2.936	1.873–4.601	0.000	2.805	1.970–3.994	0.000	2.458	1.709–3.535	0.000
Log (PSAD)	22.300	6.809–73.042	0.000				72.227	16.817–310.206	0.000	9.758	2.458–39.220	0.001

TPSA: total prostate-specific antigen; fPSA: free prostate-specific antigen; P2PSA: (-2)pro-prostate-specific antigen; PHI: prostate health index; f/T: free/total prostate-specific antigen; %P2PSA: defined as [(P2PSA/fPSA) × 100]; PV: prostate volume; PI-RADS: Prostate Imaging-Reporting and Data System; PSAD: prostate-specific antigen density; PCa: prostate cancer; OR: odds ratio; CI: confidence interval. CSPCa: clinically significant prostate cancer, defined as Gleason Grade ≥ 2.

**Table 5 jcm-12-00339-t005:** Detailed information about the predictive model used to calculate the probability of PCa.

Risk Factors	Coefficient	SE	OR (95% CI)	*p*
PCa				
Intercept	−8.508	1.754	0.000	0.000
Age	0.058	0.024	0.970 (0.952–0.988)	0.014
PHI	0.036	0.008	1.037 (1.022–1.052)	0.000
PV	−0.030	0.010	0.970 (0.952–0.988)	0.002
PI-RADS	1.077	0.229	2.936 (1.873–4.601)	0.000
CSPCa				
Intercept	−5.341	1.717	0.005	0.002
Age	0.020	0.023	1.020 (0.975–1.067)	0.383
PHI	0.032	0.007	1.032 (1.018–1.047)	0.000
PI-RADS	0.850	0.217	2.340 (1.529–3.580)	0.000
Log (PASD)	2.515	0.835	12.370 (2.406–63.583)	0.003

PCa: prostate cancer; CSPCa: clinically significant prostate cancer, defined as Gleason Grade ≥ 2; SE: standard error; OR: odds ratio; CI: confidence interval. Probability of PCa in PSA 4–20 ng/mL can be calculated by using the following formula: ln (p/1-p) = 0.058 × Age + 0.036 × PHI-0.030 × PV + 1.077 × PI-RADS-8.508. Probability of CSPCa in PSA 4–20 ng/mL can be calculated by using the following formula: ln (p/1-p) = 0.020 × Age + 0.032 × PHI + 0.850 × PI-RADS + 2.515 × Log (PSAD)-5.341.

**Table 6 jcm-12-00339-t006:** Predictive performance of different cut-off values of PHI and optimal cut-off values of nomograms.

	Sensitivity	Specificity	PPV	NPV	% Biopsy Avoided	% PCa Missed	%CSPCa Missed
PHI ≥ 35	95.77	34.90	41.21	94.55	23.64	1.36	1.36
PHI ≥ 40	90.14	45.64	44.14	90.67	30.91	3.18	1.82
PHI ≥ 45	81.69	59.73	49.15	87.25	40.45	5.91	3.18
PHI ≥ 50	76.06	71.81	56.25	86.29	48.64	7.73	4.09
PHI ≥ 55	74.65	79.87	63.86	86.86	54.09	8.18	4.55
^a^ NP ≥ 27%	88.73	82.55	70.79	93.89	55.91	3.67	1.82
^b^ NP ≥ 31%	83.64	89.09	71.88	94.23	63.64	7.27	4.09

NP: nomogram predictive; a: nomogram for predicting PCa; b: nomogram for predicting CSPCa; PPV: positive predictive value; NPV: negative predictive value; PCa: prostate cancer; CSPCa: clinically significant prostate cancer, defined as Gleason Grade ≥ 2.

## Data Availability

The datasets analyzed during the current study are available in the QILU Hospital of Shandong University.

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
