# Peer review of "Nomograms Combining PHI and PI-RADS in Detecting Prostate Cancer: A Multicenter Prospective Study"

_jcm, 2023, doi:10.3390/jcm12010339_

Round 1
Reviewer 1 Report
Report on “Nomograms combining PHI and PI-RADS in detecting prostate cancer: a multicenter prospective study” by Yongheng Zhou et al.
The manuscript proposes a new nomogram in the Asian population to combine phi-score and PI-RADS report to predict PCa or CsPCa. Nomograms are welcome, especially if an app is provided. Most of the manuscript is clear and the topic is current and interesting.
My main concerns are the following:
11) 2.1 Study population: The population comes from 3 different hospitals, is interesting to perform a comparison between them on the main variables of the study. A descriptive analysis and the Kruskall Wallis can be applied for this purpose.
22) 2.3 “The factors with P value less than 0.05 and some factors (such as age) with clinically recognized were included into the multivariable logistic regression”. I think this is too restrictive, it is usual to select as candidate variables with a univariate p-value <0.2.
33) The split of the database in training and validation cohort to validate the multivariate model was not described in material and methods. Please complete.
44) 2.5: The statistical analysis is not well described: “The normality test was first performed on the continuous variables, and the Student t-test was used for continuous variables that met the normality test, otherwise the Mann-Whitney U test was applied.” These tests were used to compare groups, please describe well. The sentence “Ranked data were analyzed by using the Mann–Whitney U test.” doesn't make any sense, the Mann-Whitney is a ranked test, but it is not used for ranked data. Please remove it. In addition, chi-square test to compare categorical variables was not described.
55) 3.1. Univariable and Multivariable regression analyses in predicting PCa and CSPCa . This paragraph has clear flaws.
Univariable analysis is not informative of independent factors, only multivariate analysis.
“Multivariable regression analysis revealed that age…were predictive factors in detecting PCa” This is wrong, only factors with p-value <0.05 can be considered predicted factors. Please correct.
66) 3.2 The construction and performance of nomogram. PSAD was transformed using a log function, but non-linear dependence of PSAD was explored using splines? Please inform about it.
77) I suppose that the Optimal cut-off point was estimated using the Youden index, please inform in the manuscript how was chosen.
88) I congratulate the authors on the app developed with shiny, perhaps in the model summary it would be good to incorporate the AUC and the sensitivity and specificity for the optimal cut-off point.
Author Response
Response to Reviewer 1 Comments
Point 1: 2.1 Study population: The population comes from 3 different hospitals, is interesting to perform a comparison between them on the main variables of the study. A descriptive analysis and the Kruskall Wallis can be applied for this purpose.
Response 1: Thank you very much for your comments. Our comparison of the three hospitals at baseline is presented in Table S1 and described in rows 149-150 of the text.
Point 2: 2.3 “The factors with P value less than 0.05 and some factors (such as age) with clinically recognized were included into the multivariable logistic regression”. I think this is too restrictive, it is usual to select as candidate variables with a univariate p-value <0.2
Response 2: Thanks for your comments. We rewrote rows 116-117 in the text to include variables with p-values less than 0.1 in univariable logistic regression analysis into the multivariable logistic regression analysis. We believe that this restricted scope may be more applicable to the study in this paper. We also checked the Tables. Thank you very much for your comments!
Point 3: The split of the database in training and validation cohort to validate the multivariate model was not described in material and methods. Please complete.
Response 3: We are very sorry for our negligence of not describing the training and validation cohort in the text. We have changed line 113-115 in the text.
Point 4: 2.5: The statistical analysis is not well described: “The normality test was first performed on the continuous variables, and the Student t-test was used for continuous variables that met the normality test, otherwise the Mann-Whitney U test was applied.” These tests were used to compare groups, please describe well. The sentence “Ranked data were analyzed by using the Mann–Whitney U test.” doesn't make any sense, the Mann-Whitney is a ranked test, but it is not used for ranked data. Please remove it. In addition, chi-square test to compare categorical variables was not described.
Response 4: We replaced the word " The normality test was first performed on the continuous variables, and the Student t-test was used for continuous variables that met the normality test, otherwise the Mann-Whitney U test was applied." with " For the comparison of the continuous variables of groups, the normality test was first performed, and the Student t-test was used for continuous variables that met the normality test, otherwise the Mann-Whitney U test was applied for continuous variables." shown in line 136-137 of the text. In this paper, we did not study categorical variables, so we did not add this sentence “chi-square test was used to compare categorical variables”. We changed this sentence “Ranked data were analyzed by using the Mann–Whitney U test.” into “Ranked data were analyzed by using the Wilcoxon rank sum test” in line 141 of the text. Thank you very much for your comments!
Point 5: 3.1. Univariable and Multivariable regression analyses in predicting PCa and CSPCa. This paragraph has clear flaws.
Univariable analysis is not informative of independent factors, only multivariate analysis.
“Multivariable regression analysis revealed that age…were predictive factors in detecting PCa” This is wrong, only factors with p-value <0.05 can be considered predicted factors. Please correct.
Response 5: Thank you for your good comments. We re-stated the clinical indicators with p-values less than 0.1 for univariable regression analysis as risk factors in line 181-182 of the text. We used multivariable stepwise forward regression analysis to describe the independent predictors in line 185-195 of the text.
Point 6: 3.2 The construction and performance of nomogram. PSAD was transformed using a log function, but non-linear dependence of PSAD was explored using splines? Please inform about it.
Response 6: We performed a linear analysis of PSAD as well as age, PHI, PV, and PI-RADS in detecting PCa. The covariance statistics VIF of all indicators are less than 3. No covariance problems were found. We then included PSAD as well as other factors in the multivariable regression analysis and we found a P value for PSAD = 0.111, OR = 8.42 (95% CI: 0.612-115.803). For over-inflated OR values, we decided to use the logarithmic transformation. The log (PSAD) was subsequently included in the multivariable stepwise forward regression analysis and the nomograms were constructed.
Point 7: I suppose that the Optimal cut-off point was estimated using the Youden index, please inform in the manuscript how was chosen.
Response 7: Thank you for your comments. The Optimal cut-off point was estimated using the Youden index in this manuscript and we have added this sentence in line 143-144 of the text.
Point 8: I congratulate the authors on the app developed with shiny, perhaps in the model summary it would be good to incorporate the AUC and the sensitivity and specificity for the optimal cut-off point.
Response 8: Your comments are very beneficial to us. But unfortunately, the model summary in the shiny website does not support us to incorporate the AUC and the sensitivity and specificity for the optimal cut-off point. Your comments are very useful to us as we plan to develop other nomograms and hope to incorporate them into the app we are developing in the future. My sincere thanks once again.
Special thanks to you for your good comments.
Reviewer 2 Report
Thank you very much for the opportunity to review this manuscript.
The authors, based on the prospectively collected data of 293 patients who underwent biopsy of the prostate, developed a nomogram that could help in determining the risk of clinically significant prostate cancer diagnosis. I believe that the results are of great clinical significance and utility, as still we have no ideal tool that would help in most accurate selection of patients who should undergo biopsy of the prostate. The nomogram has a reasonable chance to become routinely used in clinical practice, after, of course, having been externally validated.
The manuscript is well written, with cery minor language or stylistic errors. In the Introduction, the problem of low specifity of available tools in terms of predicting csPC is presented briefly and accurately. Methods are presented clearly, however, some minor issues can be pointed out and I have listed them below. The results are demonstrated properly, with details in regard to the process of the development of the nomogram being explained very thoroughly, which I consider a strength of the paper. The Discussion is well constructer and covers the majority of the topics that should be addressed. The authors are aware of the limitations of the study.
Below I have listed my comments to the manuscript:
1. The statement you provide in line 34 (suggesting an impact of PSA screening on reduction in PCa mortality) is very controversial, please revise it to be less judgmental. Moreover, there is much more new evidence than the 2008 paper you refer to.
2. In line 70 you introduce abbreviations which are not explained this first time they appear.
3. Line 75: Please specify how were the gland diameters taken - with TRUS or MRI?
4. Please specify, whether the biopsy was software fusion biopsy or cognitive biopsy? Who performed the biopsies and what was their experience? In cases of transperineal biopsy, was it free hand or template?
5. Interestingly, more than one third of your patients were assigned PIRADS category 2. Such patients are not typically considered candidates for biopsy, unless other clinical features, suggestive of clinically significant cancer, are present, e.g., markedly elevated PSA or positive DRE. All of your patients had negative DRE. So, what was the reason to perform biopsy in those 117 (39.9%) patients? Did you have any specific criteria for why a biopsy was offered to your patients? In the methods section you wrote that PSA > 4 ng/mL was an inclusion criterion to the study. Did you consider a man with a single PSA level of 4.1 ng/mL eligible for biopsy even if mpMRI showed no suspicious lesion? Please revise the methods section so that this would be clear to the readers. Also, if that was true, please discuss that you might have offered biopsy to multiple men who would not be considered candidates for biopsy on a regular basis, which may pose a risk of selection bias.
6. In regard to Table 3. I am not a statistician, but a urologist, so please explain, whether including TPSA, PHI, and f/T, which are variables dependent on each other, into a multivariable model is methodologically correct? Same with PV, TPSA and Log(PSAD).
7. Line 200: Should not criteria for optimal cut-off be first defined? What did you consider as optimal? Some may aim for optimal sensitivity, some for specificity, some for optimal sensitivity: specificity ratio. Usually, the latter is used as criterion, but, please, explain.
8. The sentence that ends in Line 249 would look better with a reference. Same with the sentence that starts in Line 254. Words “many” or “it is well known” should be accompanied with a reference.
9. In Line 287 you say that "age has been clinically identified as a risk factor". This should be accompanied with a reference.
10. In Line 290 you state that “PHI is gradually becoming routine” - please provide a reference. My observations demonstrate that the situation is rather opposite. PHI testing is relatively expensive and not easily accessible in many communities across the world.
11. Given the above issue, please discuss whether your nomogram would be applicable in a typical clinical scenario, in which a patient may not have access to PHI testing. Obviously, it may largely depend on insurance coverage or other socioeconomic aspects.
12. Line 307: Please add a reference to the statement that the cut-off value of 35 is commonly used.
13. Line 317: I believe that the prevalence of maximum core length being considered a criterion for csPCa is not high enough to be considered as “often”, please revise or add reference.
14. What your study was aimed at is not providing a method to assess the risk of harboring csPC, but risk of being diagnosed with csPC at biopsy. Of course, it is the biopsy result which determines further management, so ability to predict the biopsy result in order to predict whether the patient would proceed to active treatment or not is important, if not to say - the most important, in the context of the dilemma of offering a biopsy or not. However, referring the results of your nomogram to the risk of csPC at radical prostatectomy or risk of adverse pathologic features at radical prostatectomy would also be of great clinical significance. I would not this a limitation to your study. Rather, a direction for future research. Please, discuss it.
15. In Line 322, most probably you meant “the combined use”. Moreover, there is more than just PHI and PI-RADS in the nomogram, so please list all the factors. Same applies to the Abstract.
16. The statement that "the visualization of nomograms is more conducive to promotion of clinical application" is not supported by the data. I mean, this is not a conclusion to the results.
In summary, I recommend this manuscript for publication after the abovementioned queries are addressed. the number of issues is significant.
It was a pleasure to review this article.
Author Response
Response to Reviewer 2 Comments
Point 1: The statement you provide in line 34 (suggesting an impact of PSA screening on reduction in PCa mortality) is very controversial, please revise it to be less judgmental. Moreover, there is much more new evidence than the 2008 paper you refer to.
Response 1: We have revised the statement in the text and cited new literature, as shown in line 35-36. Your comments are greatly appreciated.
Point 2: In line 70 you introduce abbreviations which are not explained this first time they appear.
Response 2: We have restated the phrases that were not described in the first abbreviation of the article in line 78.
Point 3: Line 75: Please specify how were the gland diameters taken - with TRUS or MRI?
Response 3: We have restated this statement and labeled the gland diameters as measured by MRI in line 86 of the text.
Point 4: Please specify, whether the biopsy was software fusion biopsy or cognitive biopsy? Who performed the biopsies and what was their experience? In cases of transperineal biopsy, was it free hand or template?
Response 4: We elaborate on these questions and make additional clarifications in lines 105-108 of the text. Thanks for your comments.
Point 5: Interestingly, more than one third of your patients were assigned PIRADS category 2. Such patients are not typically considered candidates for biopsy, unless other clinical features, suggestive of clinically significant cancer, are present, e.g., markedly elevated PSA or positive DRE. All of your patients had negative DRE. So, what was the reason to perform biopsy in those 117 (39.9%) patients? Did you have any specific criteria for why a biopsy was offered to your patients? In the methods section you wrote that PSA > 4 ng/mL was an inclusion criterion to the study. Did you consider a man with a single PSA level of 4.1 ng/mL eligible for biopsy even if mpMRI showed no suspicious lesion? Please revise the methods section so that this would be clear to the readers. Also, if that was true, please discuss that you might have offered biopsy to multiple men who would not be considered candidates for biopsy on a regular basis, which may pose a risk of selection bias.
Response 5: First of all, we thank you very much for this comment. For patients in PI-RADS category 2, some patients were included in this study because of markedly elevated PSA levels or elevated PHI levels. As you asked, there is a small population with MRI suggestive of less likely cancer and PSA may be around 4-5 ng/ml. We will be elaborating different treatment options for this population, including options such as active surveillance. However, there are some patients who have a strong desire for biopsy even though we have informed them of the complications associated with biopsy, and these patients will be included in this study with signed informed consent. You mentioned that if a patient's mpMRI does not show a suspicious lesion and the PSA level is 4.1 ng/ml, we do not recommend that he undergo a biopsy, but instead we recommend that he undergo active monitoring, but if the patient chooses to undergo a biopsy despite our advice, we will respect the patient's opinion. This was not the case in all 117 patients with PI-RADS 2. This situation is also discussed in our paper in the method in line 67-70 of the text.
Point 6: In regard to Table 3. I am not a statistician, but a urologist, so please explain, whether including TPSA, PHI, and f/T, which are variables dependent on each other, into a multivariable model is methodologically correct? Same with PV, TPSA and Log(PSAD).
Response 6: It is a great pleasure to answer this question of yours. Using univariable logistic regression analysis, we identified some risk factors that could predict PCa or CSPCa. Using SPSS software, we included them in a multivariable stepwise forward regression analysis, a statistical analysis that automatically identifies the presence of covariance between each variable and incorporates the best variables. We analyzed the included variables from a clinical perspective and made the final decision to include these variables. Cointegration between variables such as PV, TPSA and Log(PSAD) is identified by the software and incorporated into the best multivariable model improvement.
Point 7: Line 200: Should not criteria for optimal cut-off be first defined? What did you consider as optimal? Some may aim for optimal sensitivity, some for specificity, some for optimal sensitivity: specificity ratio. Usually, the latter is used as criterion, but, please, explain.
Response 7: Thank you very much for your comments. We have reformulated the methodology in line 143-144 of the text and the optimal cut-off value is defined in the text as obtained by the maximum Youden index.
Point 8: The sentence that ends in Line 249 would look better with a reference. Same with the sentence that starts in Line 254. Words “many” or “it is well known” should be accompanied with a reference.
Response 8: We have added a reference to the sentence you mentioned in the text at line 249, and it is shown in the revised article at line 270. The previous text already had a reference in that sentence in lines 252-254, so we did not reinsert the reference. We have also added a reference in the revised article at line 278.
Point 9: In Line 287 you say that "age has been clinically identified as a risk factor". This should be accompanied with a reference.
Response 9: We added the reference in line 310 of the revised article. Thank you for your comments.
Point 10: In Line 290 you state that “PHI is gradually becoming routine” - please provide a reference. My observations demonstrate that the situation is rather opposite. PHI testing is relatively expensive and not easily accessible in many communities across the world.
Response 10: We have rephrased the sentence and added the reference to line 313 in the text.
Point 11: Given the above issue, please discuss whether your nomogram would be applicable in a typical clinical scenario, in which a patient may not have access to PHI testing. Obviously, it may largely depend on insurance coverage or other socioeconomic aspects.
Response 11: We discuss this issue that you raised in lines 314-317 of the text.
Point 12: Line 307: Please add a reference to the statement that the cut-off value of 35 is commonly used
Response 12: We have added a reference in line 334 of the revised text.
Point 13: Line 317: I believe that the prevalence of maximum core length being considered a criterion for csPCa is not high enough to be considered as “often”, please revise or add reference.
Response 13: We have reworked this statement to show up in line 344 of the text.
Point 14: What your study was aimed at is not providing a method to assess the risk of harboring csPC, but risk of being diagnosed with csPC at biopsy. Of course, it is the biopsy result which determines further management, so ability to predict the biopsy result in order to predict whether the patient would proceed to active treatment or not is important, if not to say - the most important, in the context of the dilemma of offering a biopsy or not. However, referring the results of your nomogram to the risk of csPC at radical prostatectomy or risk of adverse pathologic features at radical prostatectomy would also be of great clinical significance. I would not this a limitation to your study. Rather, a direction for future research. Please, discuss it.
Response 14: Many thanks to the reviewer for your comments. We discussed in the text in lines 346-351. This is a good guide for our future research direction.
Point 15: In Line 322, most probably you meant “the combined use”. Moreover, there is more than just PHI and PI-RADS in the nomogram, so please list all the factors. Same applies to the Abstract.
Response 15: We have listed all the factors in the nomograms in line 355-357 of the revised article. We also added specific risk factors to the conclusion of the abstract.
Point 16: The statement that "the visualization of nomograms is more conducive to promotion of clinical application" is not supported by the data. I mean, this is not a conclusion to the results.
Response 16: We have removed this statement from the text.
Special thanks to you for your good comments.

Round 2
Reviewer 1 Report
All my concerns have been addressed by the authors